

# Bioconversion of mango peels into itaconic acid through submerged fermentation and statistical optimization of parameters through response surface methodology

Shagufta Saeed[1], Sibtain Ahmed[2], Fatima Qureshi[1], Muhammad Sheraz Yasin[1], Rida Waseem[1] and Tahir Mehmood[3]

[1] University of Veterinary and Animal Sciences, Lahore, Punjab, Pakistan
[2] Bahauddin Zakariya University, Multan, Pakistan
[3] University of the Punjab, Lahore, Punjab, Pakistan

## ABSTRACT

Itaconic acid is an industrially crucial organic acid due to its broad range of applications. The main hurdle in itaconic acid production is the high cost of the substrate, *i.e.*, pure glucose, required for the fermentation process. Pakistan annually produces about 1.7 to 1.8 million metric tonnes of mango fruit. Keeping this in view, the potential of a sugar-rich fruit by-product, *i.e.*, mango peels, was analyzed to be used as a substrate for the biosynthesis of itaconic acid using *Aspergillus niger* by submerged fermentation. Different physicochemical parameters (incubation period, temperature, agitation rate, inoculum size, and pH) were optimized using the central composite design (CCD) design of response surface methodology (RSM). The maximum production of itaconic acid, *i.e.*, 4.6 g/L, was analyzed using 10% mango peels w/v (water hydrolysate), 3 mL inoculum volume after 5 days of fermentation period at pH 3, and a temperature of 32 °C when the media was kept at a 200-rpm agitation speed. The itaconic acid extraction from mango peels was done using the solvent extraction method using n-butanol. The identification and quantification of itaconic acid produced in the study were done using the Fourier Transform Infrared Spectroscopy (FTIR) spectrum and the High-Performance Liquid Chromatography (HPLC) method. According to HPLC analysis, 98.74% purity of itaconic acid was obtained in the research. Hence, it is concluded from the results that sugar-rich mango peels can act as a promising substrate for the biosynthesis of itaconic acid. Further conditions can be optimized at the bioreactor level to meet industrial requirements.

## INTRODUCTION

Organic acids are used by many companies, especially in the manufacturing industry. These comprise lactic, citric, gluconic, and itaconic acids that have undergoing extensive bioprocessing. The most noteworthy of these is itaconic acid (IA). This white, crystalline carboxylic acid is created during the fermentation of polysaccharides (*Ajiboye et al., 2018*).

Corresponding authors
Sibtain Ahmed, sibtain@bzu.edu.pk
Tahir Mehmood, tahir.mmg@pu.edu.pk

IA was initially identified by Baup in 1836 as the byproduct of citric acid distillation. IA a derivative of an unsaturated carboxylic acid, is recognized as a platform chemical due to its various applications. IA manufacturing costs have risen to fulfill industrial demand due to the scarcity of fossil resources. Additionally, it is generated by fungi, such as *Aspergillus* spp. and *Candida* spp. It can, therefore, be generated biosynthetically to cut costs and meet industrial demands (*Gnanasekaran et al., 2018*).

The worldwide IA market was approximately $75 million in 2015. By 2021, it had climbed to $95.4 million, and projections suggest it will reach about $108.4 million by 2026 (*Gopaliya, Kumar & Khare, 2021*). IA is produced mainly in China, Japan, the United States, and France (*Weastra, 2013*).

IA serves as a superabsorbent and helps remove phosphate from detergents, indicating its industrial significance. Additionally, it is used to make methyl tetrahydrofuran, a biofuel. Nitrilon is an IA product utilized in the paper industry and water treatment systems (*Saha et al., 2017*).

IA, made from biomaterials is an adequate substitute for acrylic Acid derived from petrochemicals. Trifunctionality enables the synthesis of several novel polymers with applications in coatings, elastomers, smart nano hydrogels for food applications, and targeted drug administration (particularly in the treatment of cancer) using IA and its derivatives (*Teleky & Vodnar, 2021*). IA is used to develop biodegradable food packaging materials by combining it with other polymers to create environmentally friendly alternatives to traditional plastic packaging. In the referenced study, IA was combined with poly (vinyl alcohol) (PVA) and pectin to produce biodegradable and water-soluble packaging materials. These materials were further enhanced by incorporating organic or phenolic extracts from apple by-products, which possess significant antioxidant activity. The incorporation of IA and these extracts not only improved the antioxidant properties but also influenced the physical-chemical properties of the biofilm solutions, such as viscosity and water vapor permeability, making them suitable for use in the food industry (*Teleky et al., 2022*).

*Aspergillus* species naturally produce IA. The subspecies of *Aspergillus producing* itaconic acid include *Aspergillus niger*, *Aspergillus niveus*, *Aspergillus flavus*, and *Aspergillus terreus*. *Aspergillus* species are the primary species for the hyper-production of IA under phosphate-deficient conditions but rely more on glucose nutrition. When exposed to a substrate, other microbes, including *Candida* species, *Escherichia coli*, *Rhodotorula*, *Yarrowia lipolytica*, *Ustilago maydis*, and *Ustilago zeae*, have been reported to synthesize IA (*Hegde et al., 2016*).

Using genetic engineering technique make IA can be genetically altered. The yield of IA production increases in another fungal species if the cadA gene is expressed in that species (*Li et al., 2011*). Various microbes have used glucose as a medium during fermentation to produce IA. Fermentation in solid or submerged states is an option. However, this approach using glucose as a substrate is not economical. Various low-cost substrates are being investigated to produce IA at a cheap cost (*Bafana & Pandey, 2018*).

According to the Food and Agriculture Organization, fruits, vegetables, and root crops account for between 40–50% of the global food waste. Only fruit and vegetable waste

accounts for 37% of total agricultural waste in Asia. FAO estimates that 35 to 40 percent of all fruits and vegetables are wasted annually. Increasing fruit consumption, fruit storage, and commercial fruit processing all result in increasing fruit waste. Managing this garbage is also troublesome (*Save food, 2015*; *Zhu et al., 2023*).

The development of submerged fermentation techniques has improved bioprocesses, such as the biodegradation of hazardous materials and their bioremediation. These techniques can produce added-value substances, including enzymes, organic acids, antibiotics, plant growth factors, enzymes, biopesticides, biofuel, andthe biotransformation of crops into very nutrient-dense foods (*Jahid, Gupta & Sharma, 2018*).

Agro-waste can potentially cut IA prices because it is a less expensive substrate than glucose and sucrose (*Devi et al., 2022*). This organic acid is produced by jatropha seed cake, corn syrup, and molasses, IA can also be made using the starches of sorghum, wheat, sago, sweet potatoes, cassava, and potatoes (*El-Imam & Du, 2014*). For instance, low-cost substrate sources include sugarcane bagasse, rice husk, rice bran, crushed nuts, tamarind seeds, orange peels, and orange pulp through solid state fermentation (*Rafi et al., 2014*).

The current study investigates the feasibility of using mango peels, a cheap and sugar-rich fruit by-product, as a substrate for producing IA by altering many physicochemical parameters through submerged fermentation and production parameters optimized by RSM. It can be further tuned to the bioreactor level to meet industry demands.

# MATERIALS AND METHODS

## Chemicals

1) Sabouraud dextrose agar 2) NaCl 3) Tween 80 4) $H_2SO_4$ 5) Bromine reagent: bromine, potassium chloride, potassium bromide, hydrochloric acid 6) itaconic acid 7) n-butanol 8) sulfuric acid 9) Anthrone reagent 10) DNS (3,5-Dinitrosalicylic acid) reagent. All the chemicals were acquired from Sigma Aldrich.

## Collection and maintenance of fungal strain

The fungal strain of *Aspergillus niger* (FCBP-PTF-684) was obtained from the University of the Punjab's first Fungal Culture Bank, Lahore, Pakistan. The microorganism was revived on Sabouraud dextrose agar plates. The appearance of microbial growth, the plates were preserved in the refrigerator at 4 °C (*Saeed et al., 2021*).

## Inoculum preparation

A loop full of spores of *A. niger* was mixed well in the sterilized 0.9% NaCl solution with 0.1% Tween 80. The spore suspension with $3 \times 10^7$ spores/mL was used as an inoculum for further research (*Martău et al., 2021*).

## Collection of substrate

Mango (*Mangifera indica*) peels were collected from the University of Veterinary and Animal sciences. The mango peels were washed properly and dried in a hot air oven at 40 °C overnight. Then, they were grounded into the fine powder form and sieved through

the mesh of particle size 0.5 mm. The powder was stored in a polythene zip bag at room temperature (*Saeed et al., 2020*).

## Preparation of mango peel hydrolysates

Acid and water hydrolysates of mango peels were prepared as follows: For acid hydrolysis 2, 4, 6, 8, 10 and 12 g of mango peels powder were mixed with 100 mL of 0.05 N $H_2SO_4$ solution. The flasks were then incubated at 50 °C for 60 min in the shaking incubator at 150 rpm. The hydrolysates were filtered through a thin membrane filter paper of 0.2 μm. The filtrates were further diluted up to 100 mL by adding distilled water and autoclaved. Water hydrolysates were prepared by adding 2, 4, 6, 8, 10, and 12 g of mango peel powder to 100 mL distilled water and autoclaved at 121 °C for 15 min, then filtered through filter paper. The total sugar content of the mango peel hydrolysates (acid and aqueous) was estimated using the Anthrone method. Reducing sugars were estimated by the DNS (3,5-Dinitrosalicylic acid) method (*Arumugam & Manikandan, 2011*).

## Selection of best substrate for itaconic acid production

The ideal substrate for IA synthesis was selected by using 25 mL of each mango peel hydrolysate (acidic and aqueous) as substrate in 250 mL conical flasks. Glucose solutions (2%, 4%, 6%, 8%, 10%, and 12%) were also prepared as a control to compare and select the substrate that will give high IA. The pH of the fermentation media was set at seven and autoclaved. Then 2 mL *Aspergillus niger* inoculum was put into each flask. The flasks were then incubated at 30 °C on the rotary shaker for 120 h and 150 rpm of agitation speed. The hydrolysate that produced the highest amount of IA was chosen to optimize cultural parameters.

## Optimization of parameters for maximum production of itaconic acid

Optimization of parameters for achieving high yield of IA was done using central composite design (CCD) of response surface methodology (RSM). The optimized parameters are temperature 20 °C to 40 °C, inoculum size 1 to 5 mL, pH 3 to 8, agitation rate 100 to 300 rpm, and incubation period 1 to 7 days (*Kerssemakers et al., 2020*), with the software Minitab 17, a table was prepared to optimize IA production. A total of 32 tests were done with different parameters to indicate the interaction between them and to verify their combined and individual effect on the production of IA (*Ahmad et al., 2013*). One-way ANOVA analyzed the effect of parameters on IA. If the *p*-value is less than 0.05, then the effect of the factor is considered as significant.

The estimation of IA was done using a spectrophotometric method using a bromine reagent. The contents of the bromine reagent were: 1 mL bromine, potassium chloride 1.87 g, potassium bromide 3 g, 1N hydrochloric acid 48.50 mL, and water 500 mL. It was then stored in an amber bottle. In 3 mL cuvette, 0.3 mL of bromine reagent was added and made its volume up to 1 mL by adding distilled water. Then, HCl at pH 1.2 was added up to 3 mL and left it for 15 min. This solution served as the blank at 385 nm. In another cuvette, 0.3 mL bromine reagent and 1 mL standard of IA were added, and volume was made up to 3 mL by using HCl, and then optical density was measured. The same procedure was

repeated with a sample of IA produced, and the amount of IA produced was estimated (*Ajiboye et al., 2018*). Table 1 illustrates the range of different parameters tested to achieve hyperproduction of IA.

### Extraction and purification of Itaconic acid

Purification of IA through a solvent extraction process utilizing n-butanol as the solvent. The IA broth was filtered using Whatman (0.2 μm) filter discs. Subsequently, an aqueous solution of IA was prepared by dissolving it in an equal amount of deionized water. Another filtration step was carried out using a Whatman (0.2 μm) syringe filter. The saturated IA solution was then combined with the organic solvent (n-butanol) in varying ratios (1:1, 1:2, 1:3, 1:4) to determine the optimal volume of extractant for maximum purification. The solutions were thoroughly mixed for 45 min using a magnetic stirrer at 500 rpm. The resulting mixture was transferred to a 500 mL separating funnel and allowed to settle for 1 h, forming two distinct phases based on the density contrast between the aqueous and organic phases. Following phase separation, the volumes of the aqueous and organic phases were measured. Both phases were analysed to determine the concentration of IA through titration using different n-butanol ratios. The degree of extraction (%E) was subsequently calculated (*Bose, 2014*).

### Characterization of itaconic acid

IA produced was identified using the Fourier Transform Infrared Spectroscopy (FTIR) spectrum using the standard from Sigma-Aldrich. IA purity was estimated using the High-Performance Liquid Chromatography (HPLC) technique (*Gnanasekaran et al., 2018*). The cell mass concentration was determined based on the dry weight of the cells. The cell mass from the fermentation broth was collected through centrifugation at 10,000 $g$ for 10 min and washed three times with deionized water. The washed cell mass was then dried thoroughly at 80 °C until a constant weight was achieved. The fermentation broth, post-centrifugation at 10,000 $g$ for 10 min, was stored at −20 °C. IA quantification was carried out using HPLC with a column length of 0.3 m and a diameter of 7.8 mm, operating at 37 °C with 0.005 M sulfuric acid as the mobile phase at a flow rate of 20 uL min$^{-1}$. Detection was done using a UV-Vis diode array detector at 210 nm (*Bose, 2014*).

### Estimation of the carbohydrate content of mango peels hydrolysates

The Anthrone method assessed the total sugar content of the acid hydrolysate and water hydrolysate of mango peels while the DNS method was used to estimate reducing sugar content. Table 2 displays the amount of sugars present.

## RESULTS AND DISCUSSIONS

### Selection of best mango peels hydrolysate for itaconic acid production

Various glucose concentrations were tested alongside different acid and water hydrolysate concentrations of mango peels w/v to determine the best substrate for IA production (Fig. 1). The highest amount of IA (4.6 g/L) was produced from the water hydrolysate of

**Table 1 Level of independent factors in the designed experiment.**

| Codes | Independent factors | Unit | Low level | High level |
|---|---|---|---|---|
| A | Temperature | °C | 20 | 40 |
| B | Incubation period | hrs | 24 | 168 |
| C | pH | – | 3 | 8 |
| D | Inoculum size | mL | 1 | 5 |
| E | Agitation rate | rpm | 100 | 300 |

**Table 2 Estimation of carbohydrate content of hydrolysates of mango peels (g/100 mL).**

| Contents | Acid hydrolysate | Water hydrolysate |
|---|---|---|
| Total sugars | 18.50 ± 0.20 | 20.50 ± 1.50 |
| Reducing sugars | 14.21 ± 0.30 | 17.3 ± 0.50 |

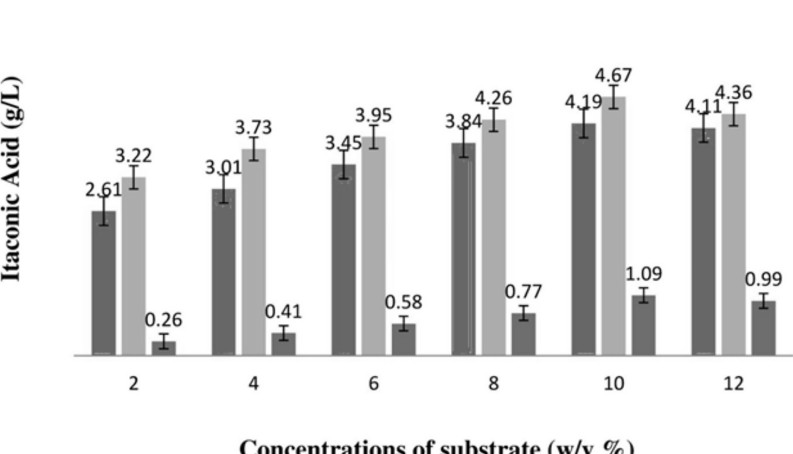

**Figure 1 Amount of itaconic acid produced by acid and water hydrolysis of mango peels using submerged fermentation.**

mango peels (at 10 w/v%), Fig. 1 shows the quantity of IA produced using various acid and water hydrolysate concentrations.

## Optimization of itaconic acid production by RSM

The ranges of different parameters that were optimized are: Temperature, 20 °C to 40 °C, inoculum size; 1 to 5 mL, pH; 3 to 8, agitation rate; 100 to 300 rpm and incubation period; 1 to 7 days The best IA (4.6 g/L) production by water hydrolysate of mango peels can be achieved at temperature of 32 °C after an incubation period of 120 h at pH 5, with inoculum size 3 mL and agitation rate of 200 rpm, as demonstrated in Table 3. The least amount of IA (2.64 g/L) was achieved after 24 h at pH 3, with an inoculum size of 1 mL and an agitation speed set at 100 rpm.

**Table 3 Central composite design for optimization of itaconic acid production from mango peels hydrolysate.**

| Sr. No. | A | B | C | D | E | Itaconic acid (g/L) | | |
| --- | --- | --- | --- | --- | --- | --- | --- | --- |
| | | | | | | Actual | Predicted | Residual |
| 1 | 35 | 72 | 5 | 3 | 200 | 3.46 | 3.49 | −0.03 |
| 2 | 30 | 48 | 3 | 2 | 100 | 2.95 | 2.97 | −0.02 |
| 3 | 20 | 24 | 4 | 1 | 300 | 2.75 | 2.76 | 0.00 |
| 4 | 30 | 144 | 7 | 4 | 150 | 4.34 | 4.36 | −0.02 |
| **5** | **32** | **120** | **5** | **3** | **200** | **4.6** | **3.97** | **0.63** |
| 6 | 40 | 24 | 8 | 5 | 300 | 3.42 | 3.41 | 0.01 |
| 7 | 30 | 72 | 6 | 3 | 100 | 3.45 | 3.46 | −0.01 |
| 8 | 20 | 48 | 3 | 1 | 150 | 2.85 | 2.86 | −0.01 |
| 9 | 30 | 96 | 4 | 5 | 250 | 3.86 | 3.87 | −0.01 |
| 10 | 35 | 120 | 5 | 4 | 100 | 3.98 | 4.01 | −0.03 |
| 11 | 32 | 144 | 7 | 2 | 150 | 4.15 | 4.19 | −0.04 |
| 12 | 25 | 168 | 6 | 4 | 250 | 4.57 | 4.59 | −0.03 |
| 13 | 30 | 96 | 4 | 5 | 200 | 3.83 | 3.84 | −0.01 |
| 14 | 20 | 72 | 8 | 1 | 300 | 3.45 | 3.45 | 0.00 |
| 15 | 25 | 168 | 3 | 5 | 150 | 4.44 | 4.47 | −0.03 |
| 16 | 32 | 144 | 5 | 4 | 250 | 4.29 | 4.33 | −0.04 |
| 17 | 30 | 120 | 6 | 3 | 200 | 3.98 | 4.01 | −0.03 |
| 18 | 40 | 24 | 8 | 2 | 150 | 3.04 | 3.06 | −0.01 |
| 19 | 20 | 72 | 2 | 5 | 300 | 3.51 | 3.51 | 0.00 |
| 20 | 35 | 48 | 7 | 1 | 200 | 3.14 | 3.17 | −0.02 |
| 21 | 32 | 144 | 5 | 4 | 250 | 4.29 | 4.33 | −0.04 |
| 22 | 30 | 168 | 3 | 3 | 300 | 4.35 | 4.41 | −0.06 |
| 23 | 20 | 72 | 4 | 5 | 150 | 3.53 | 3.52 | 0.02 |
| 24 | 32 | 48 | 6 | 4 | 300 | 3.43 | 3.43 | 0.00 |
| 25 | 40 | 24 | 8 | 2 | 200 | 3.07 | 3.08 | −0.01 |
| 26 | 32 | 120 | 6 | 5 | 100 | 4.12 | 4.14 | −0.01 |
| 27 | 25 | 48 | 4 | 2 | 150 | 3.02 | 3.03 | −0.01 |
| 28 | 32 | 168 | 7 | 3 | 250 | 4.54 | 4.59 | −0.04 |
| **29** | **20** | **24** | **3** | **1** | **200** | **2.64** | **2.65** | **−0.01** |
| 30 | 35 | 144 | 5 | 5 | 250 | 4.40 | 4.44 | −0.04 |
| 31 | 25 | 120 | 6 | 3 | 300 | 4.02 | 4.04 | −0.02 |
| 32 | 40 | 72 | 3 | 1 | 150 | 3.15 | 3.21 | −0.06 |

**Note:**
Bold text shows the maximum and minimum yield of Itaconic acid.

The ANOVA illustrated that the proposed model has an F-value of 12.49 and a *p*-value of 0.000, as shown in Table 4. The coefficient of determination ($R^2$) depicts a value of 97.39%, which confirms the accuracy and precision of the model. It is calculated that 2.61% deviation was not predicted by the design for IA production. The model depicted that the effect of factors *i.e.*, incubation period and inoculum size, respectively, on IA biosynthesis was considered significant while the effect of factors A, C and E *i.e.* temperature, pH and agitation rate, respectively, were considered as insignificant.

**Table 4 Analysis of variance for itaconic acid production from mango peels.**

| Source | DF | Adj SS | Adj MS | F-Value | P-Value |
|---|---|---|---|---|---|
| Model | 20 | 10.8460 | 0.542298 | 20.49 | 0.000 |
| Linear | 5 | 1.6533 | 0.330667 | 12.49 | 0.000 |
| A | 1 | 0.0135 | 0.013498 | 0.51 | 0.490 |
| B | 1 | 0.4861 | 0.486138 | 18.36 | 0.001 |
| C | 1 | 0.0593 | 0.059277 | 2.24 | 0.163 |
| D | 1 | 0.1322 | 0.132244 | 5.00 | 0.047 |
| E | 1 | 0.0120 | 0.011995 | 0.45 | 0.515 |
| Square | 5 | 0.1122 | 0.022440 | 0.85 | 0.544 |
| A * A | 1 | 0.0227 | 0.022688 | 0.86 | 0.374 |
| B * B | 1 | 0.0058 | 0.005818 | 0.22 | 0.648 |
| C * C | 1 | 0.0008 | 0.000806 | 0.03 | 0.865 |
| D * D | 1 | 0.0015 | 0.001470 | 0.06 | 0.818 |
| E * E | 1 | 0.0587 | 0.058746 | 2.22 | 0.164 |
| 2-Way Interaction | 10 | 0.0825 | 0.008246 | 0.31 | 0.962 |
| A * B | 1 | 0.0273 | 0.027292 | 1.03 | 0.332 |
| A * C | 1 | 0.0272 | 0.027210 | 1.03 | 0.332 |
| A * D | 1 | 0.0152 | 0.015155 | 0.57 | 0.465 |
| A * E | 1 | 0.0358 | 0.035802 | 1.35 | 0.269 |
| B * C | 1 | 0.0424 | 0.042394 | 1.60 | 0.232 |
| B * D | 1 | 0.0123 | 0.012251 | 0.46 | 0.510 |
| B * E | 1 | 0.0014 | 0.001362 | 0.05 | 0.825 |
| C * D | 1 | 0.0345 | 0.034499 | 1.30 | 0.278 |
| C * E | 1 | 0.0349 | 0.034865 | 1.32 | 0.275 |
| D * E | 1 | 0.0045 | 0.004481 | 0.17 | 0.689 |
| Error | 11 | 0.2912 | 0.026471 | | |
| Lack-of-fit | 10 | 0.2912 | 0.029118 | * | * |
| Pure error | 1 | 0.0000 | 0.000000 | | |
| Total | 31 | 11.1371 | | | |

**Note:**
* Indicates not significant.

The response regression equation from Minitab 17 software is shown as Eq. (1). This equation outlines the relationship between dependent variable and independent variables. The regression model equation is shown as:

$$
\begin{aligned}
\text{Itaconic acid (g/L)} = &-0.64 + 0.135\,A + 0.0121\,B - 0.270\,C + 0.120\,D + 0.0165\,E \\
&-0.00214\,A*A - 0.000016\,B*B + 0.0040\,C*C \\
&-0.0082\,D*D - 0.000025\,E*E + 0.000490\,A*B \\
&+0.00654\,A*C - 0.00648\,A*D - 0.000364\,A*E \\
&-0.00198\,B*C - 0.00076\,B*D - 0.000003\,B*E \\
&+0.0465\,C*D + 0.000769\,C*E + 0.000298\,D*E
\end{aligned}
\tag{1}
$$

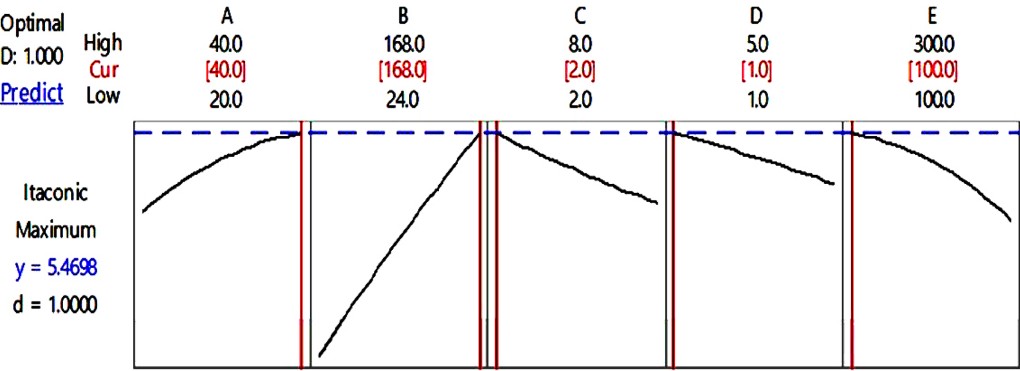

**Figure 2 Desirability chart for itaconic acid production.** Desirability chart of itaconic acid production using mango peels *via* submerged fermentation.

The desirability chart for IA production is shown in Fig. 2 which demonstrates that if the value of parameter A is 40 °C, parameter B is 168 h, parameter C is eight, parameter D is 1 mL and parameter E is 100 rpm, then the maximal predicted yield for IA synthesis would be 5.47 g/L. This response is near to expected values when compared to actual values that indicate the designed model's accurate prediction.

In Fig. 3, contour plots illustrate the various independent variables that affect IA production. These graphs show a relationship between two independent variables and a third dependent variable. The color changes exhibit the stages of IA biosynthesis that coexist among two independent variables while keeping the third component's value constant.

## FTIR analysis

The FTIR analysis was used to identify the IA produced through fermentation by using mango peel hydrolysate, and the peak pattern was compared with the standard IA obtained from Sigma Aldrich. The FTIR spectrum of the sample indicated that the produced compound is IA. IA has a peak at a wavelength of 1,629 cm$^{-1}$ for C=C, C=O at 1,704 cm$^{-1}$, and COO-H ranging from 3,600 to 2,600 cm$^{-1}$ wavenumber. The spectra were shown in Fig. 4.

## HPLC analysis

The HPLC method used, as shown in Fig. 5, reveals that IA produced in the current research is 98.74% pure compared to the standard.

## DISCUSSIONS

Pakistan's annual mango production currently stands between 1.7 and 1.8 million metric tons. Pakistan is the fifth-largest mango grower in the world (*Memon, 2016*; *Paranthaman, Kumaravel & Singaravadivel, 2014*). The sugar content of these fruit wastes can be utilized to produce useful products after fermentation like IA (*Reddy et al., 2003*). According to the findings of this study, mango peels are utilized to produce IA by *Aspergillus niger* as it has the propensity to use sugars in mango peels as a substrate in the manufacture of IA. The 10% water hydrolysate contains highest content of total sugars and reducing sugars

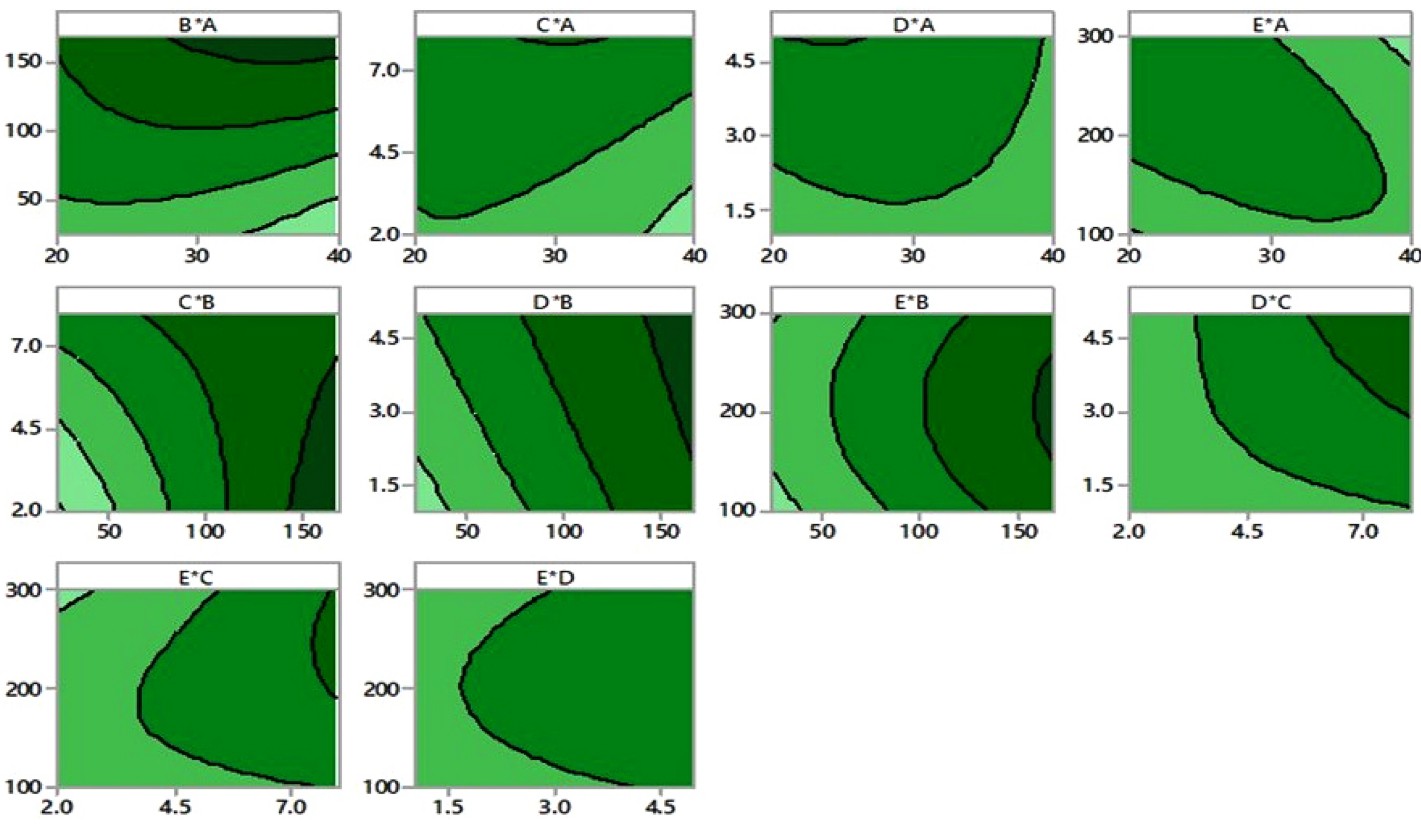

**Figure 3 Contour plots.** Contour plots of itaconic acid production showing the interaction between various factors.

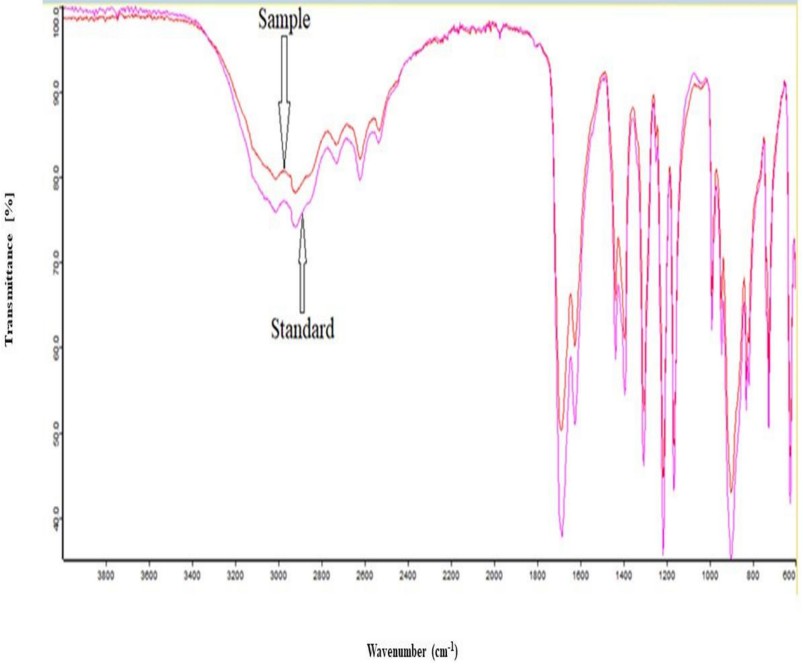

**Figure 4 FTIR analysis spectra.** FTIR analysis of itaconic acid produced by fermentation.

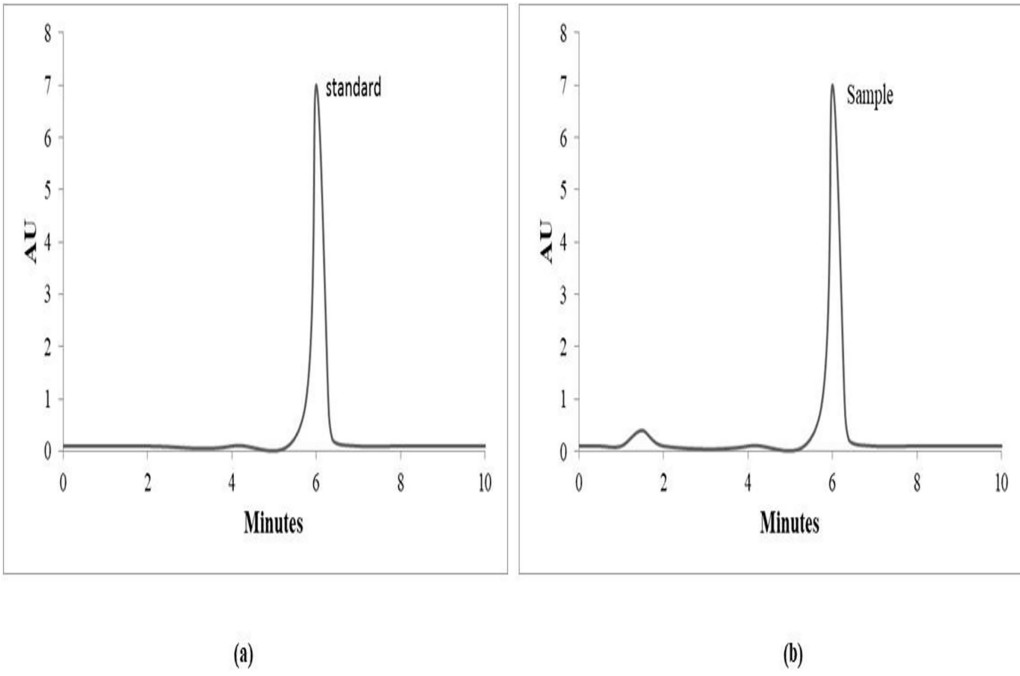

**Figure 5 HPLC analysis spectra.** HPLC chromatogram of itaconic acid (A) standard (B) sample.

(20.50 ± 1.50% & 17.30 ± 0.50%), which is in accordance with mango peels' determined carbohydrate content of another study (*Reddy, Reddy & Wee, 2011*). The high carbohydrate content of the peels serves as a sugar supply for fungus, which is engaged in the fermentation process that results in IA. Acid hydrolysate of sorghum bran was used to synthesize IA, which did not affect the growth of *A. terreus*, but the obtained yield was only one-quarter to that of pure glucose (*El-Imam & Chenyu, 2015*). In another study, the acid hydrolyzation of wheat bran was used to synthesize the IA concentration of 26.72 g/L after 144-h fermentation but with the supplementation of 100 g/L concentrated sugar (*Sun, 2016*). The research conducted by *Saha & Kennedy (2018)* in which IA production is produced from glucose, xylose, arabinose, mixed sugars, as well as dilute acid and liquid hot water pretreated wheat straw hydrolyzates. This process occurs in microtiter plate (MTP) micro bioreactors at a scale of 100 µL. There was no growth of *A. terreus* in the dilute acid hydrolysate, and no IA was produced until supplemented with 80 g/L sugar mixture (glucose, arabinose, and xylose) in 1,000-fold diluted hydrolysate. About half of the IA yield was gained after supplementing additional sugars (*Saha & Kennedy, 2018*).

Biosynthesis of IA was carried out at different temperatures ranging from 20 °C to 40 °C to observe the maximum production of IA. The maximum production of IA *i.e.* 4.6 g/L, was observed at 32 °C. For the current study, the effect of temperature on IA production was also studied using the statistical design of RSM. The maximum IA production was obtained at 32 °C, and according to ANOVA analysis, its effect remained insignificant ($p$-value > 0.05). This is because fungal cell growth and metabolism are optimum at this

temperature, and a high yield of IA was easily achieved at this temperature. On the contrary, at higher temperatures, low yield was observed because the maintenance coefficient increases at high temperatures, and with an activation energy of 15 to 20 Kcal/mol, there is a decrease in the yield coefficient. The present study is in agreement with the study that confirms the maximum production of IA (28.9 g/kg) by *Ustilago maydis* using orange pulp at 32 °C (*El-Imam & Du, 2014*). It almost agrees with the study that described that *A. terreus* utilized all sugars in the substrate (glucose, xylose, arabinose, and mixed sugars), and IA was optimally produced at 33 °C (*Rafi et al., 2014*). This is in accordance with another study which revealed that *A. niger* produces a maximum of 1.4 g/L IA by using glucose as substrate at 33 °C (*Saha et al., 2019*). In contrast to the present study, several studies reveal that the biotechnologically modified stains of *Aspergillus niger* show optimal production of IA (79 ± 0.5 g/L) at a high-temperature range of 37 °C to 40 °C (*Willke & Vorlop, 2001*).

To find the best incubation time to produce the most IA, the incubation duration was changed in the current study from 24 to 168 h. At 120 h, *A. niger* produced the highest amount of IA (4.6 g/L), according to observations. ANOVA revealed that the incubation duration had a significant impact (*p-value* < 0.05). The amount of organic acid produced in the fermentative medium directly correlates with the incubation period for organic acid synthesis. A buildup of organic acid in the fermentation media frequently prevents the growth of mushrooms. The current investigation supports the finding that IA production by various Aspergillus species peaks at 120 h (*Meena et al., 2010*). This is consistent with the study, which demonstrated that *Ustilago maydis* produced more IA with longer incubation times, reaching a maximum production at day 5. According to *Rafi et al. (2014)*, the IA yield was 28.9 g/kg. Another study confirmed this finding, reporting that *A. niger* produces the highest production of 67.67 mg/mL of IA through submerged fermentation on day 5, utilizing sweet potato peels as the fermentation's substrate (*Meena et al., 2010*). In contrast to the current findings, another study (*Gnanasekaran et al., 2018*) suggested that IA production can reach its peak after 168 h. The greatest production of IA (28.9 g/kg) from *Ustillago maydis* utilizing orange pulp was reportedly produced in 5 days, per the literature (*Rafi et al., 2014*).

Fermentation was carried out at pH ranging from 3 to 8 to optimize IA production by RSM. In the current study the highest production of IA is reported at pH 5, and its overall effect was insignificant (*p-value* > 0.05) as analyzed by ANOVA. Similar results were reported by a previous study in which parameters were optimized to produce IA. The results showed the highest production of IA at substrate concentration 4 g, inoculum size 3%, and pH 5.0 by *A. niger* and *A. terreus* to 132.9 and 157.5 g/L, respectively (*Omojasola & Adeniran, 2014*). It was in contrast with the study, which reveals that *A. niger* shows a yield of 209 g/L at pH 2.5 using jetropha seed cake and *A. terreus* gives 218 g/L at pH 3.5 (*Omojasola & Okwechime, 2017*). In another study it was reported that *Pseudozyma antarctica* NRRL Y-7808 was found to produce IA as 248 mg $L^{-1}$ $h^{-1}$ from glucose at pH 5.0 (*Levinson, Kurtzman & Kuo, 2006*). Maintaining optimal pH is important as it plays

vital role in maintaining the external and internal proton concentration for proper microbial metabolism. Microorganisms maintain their intracellular pH according to their extracellular environment in order to enhance their growth and IA production. It happens by maintaining their nutrient uptake, other physiological parameters and their metabolic pathways (*Meena et al., 2010*).

The impact of inoculum size on IA production was also examined in the current study using the statistical design of RSM. The inoculum size of 3 mL gave the highest yield of IA at 32 °C and its effect was found to be significant (*p-value* < 0.05). Microbes require optimum level of inoculum size for IA production. It is important because in case of a lower quantity of inoculum size, the fungal growth is in the lag phase and shows a reduction in the yield of IA, while the excessive inoculum will reduce the IA yield by leading the nutrient competition among microorganisms. Present study is in contrast to the study which reveals that maximum IA production by different Aspergillus species can be achieved by the 10% (v/v) inoculum size and above this concentration the production decreased due to the competition of getting nutrients (*Meena et al., 2010*). It also shows disagreement with another research work as they optimized four different parameters (incubation time, pH, inoculum size and peptone content) and got the highest yield 21.5 ± 1.87 g/kg of IA at 6% inoculum size by using *Aspergillus japonicas* from *Citrullus lanatus* (*Ramakrishnan et al., 2020*).

The current study also describes the effect of rotation speed on IA biosynthesis. Fermentation was done at different agitation rates, including 100, 150, 200, 250, and 300 rpm. It was observed the highest IA of 4.6 g/L was produced at 200 rpm and the ANOVA analysis shows the factor insignificant as *p-value* > 0.05. Ours study reported similar results to those of an earlier study, which showed that different species of Aspergillus produced the highest IA production at 200 rpm (*Meena et al., 2010*). These results agree with another study on the IA production from mixed sugars, which found that the highest IA production was achieved at 200 rpm, except for arabinose, which is fermented at 300 rpm (*Sadh, Duhan & Duhan, 2018*). Another study claims that stopping aeration will result in *A. terreus* producing no more IA. Lowering the rpms has a detrimental impact on the synthesis of IA, as demonstrated by studies utilizing various shaking speeds in flasks. While high levels of dissolved oxygen increase the production of other organic acids like citric and oxalic acid, which divert carbon away from IA production, research with *Aspergillus niger* showed that a reduced level of dissolved oxygen has a positive impact on the production of IA (*Kerssemakers et al., 2020*).

## CONCLUSION

The current study's findings suggest that mango peels can be a helpful substitute substrate for itaconic acid production. The maximum amount of itaconic acid (4.6 g/L) was produced at 32 °C, pH 5, 3 mL of inoculum size, and 200 rpm after 120 h of incubation. The biosynthesis of itaconic acid was found to be significantly impacted by the inoculum size and incubation length, but not by temperature, pH, or agitation speed. The ideal

growing conditions for the large-scale synthesis of itaconic acid can be further exploited to satisfy industrial demands.

### Funding
The authors received no funding for this work.

### Competing Interests
The authors declare that they have no competing interests.

### Author Contributions
- Shagufta Saeed conceived and designed the experiments, analyzed the data, authored or reviewed drafts of the article, and approved the final draft.
- Sibtain Ahmed analyzed the data, prepared figures and/or tables, authored or reviewed drafts of the article, and approved the final draft.
- Fatima Qureshi conceived and designed the experiments, performed the experiments, analyzed the data, prepared figures and/or tables, and approved the final draft.
- Muhammad Sheraz Yasin conceived and designed the experiments, analyzed the data, authored or reviewed drafts of the article, and approved the final draft.
- Rida Waseem conceived and designed the experiments, performed the experiments, authored or reviewed drafts of the article, and approved the final draft.
- Tahir Mehmood performed the experiments, analyzed the data, prepared figures and/or tables, authored or reviewed drafts of the article, and approved the final draft.

### Data Availability
The raw data are available in the Supplemental File.

### Supplemental Information
Supplemental information for this article can be found online at http://dx.doi.org/10.7717/peerj.18188#supplemental-information.

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
