# Peer review of "Bioconversion of mango peels into itaconic acid through submerged fermentation and statistical optimization of parameters through response surface methodology"

_PeerJ, doi:10.7717/peerj.18188_

## Round 0.1 · original submission · Major Revisions

Please address concerns of both reviewers and amend manuscript accordingly.

Reviewer 1 ·

Basic reporting

please find specific comments in the attached document.

Experimental design

please find specific comments in the attached document.

Validity of the findings

please find specific comments in the attached document.

Annotated reviews are not available for download in order to protect the identity of reviewers who chose to remain anonymous.

Reviewer 2 ·

Basic reporting

In this study, the authors report biosynthesis of itaconic acid using Aspergillus niger by submerged fermentation with mango peels as the substrate. Despite the authors' commendable efforts, the manuscript is not recommended for publication due to issues including unclear presentation of innovation, inconsistent content organization, language deficiencies leading to confusion, and the feedback provided below.

1. Title: The manuscript title is too long and utilizes incorrect terminology. Mango peels are food waste, not biowaste. If only one type of food waste is being studied, then the authors can consider stating mango peels as such without parentheses.
2. Abstract:
a. Authors must specify if the 10% mango peels are on a volume, weight, or weight/volume basis.
b. Line 27-29: FTIR and HPLC are incorrectly capitalized.
3. Introduction
a. Line 49-50: Authors should revise the predicted itaconic acid global market potential and yield from 2020 to reflect recent data.
b. Line 61: The abbreviation of itaconic acid should be defined in the beginning and utilized consistently throughout the manuscript.
c. Line 73, 88: The antecedent "this" is unclear regarding what is referred to.
d. Line 76-80, 87-88, 231-232: References are missing.
e. The authors do not clearly state the knowledge gap their study addresses and explain their choice of mango peels over other fruits and vegetables. The study would benefit from a clearer discussion of how existing literature informed the selection of feedstock, methods, and parameters.

Experimental design

4. Materials and Methods
a. Why were the mango peels dried at 80 °C for 2 hours? Usually, overnight drying at 40-70 °C is recommended and widely applied in the literature. Thus, references and explanations for the utilized method are required.
b. Line 111 and 124; 169-171: The complete acronym for UVAS and DNS should be spelled out. FTIR and HPLC are not spelled out in the manuscript. The acronym "HPLC" is spelled out for the first time on line 176 after being used in line 170.
c. Line 162: At what temperature was the extraction conducted?

Validity of the findings

5. Results and Discussion
a. The authors don’t discuss the selection of parameter ranges for optimizing itaconic acid production.
b. Why do temperature, pH, and agitation rate not significantly affect itaconic acid production? The authors should elucidate the physical significance of the ANOVA results, explain their practical implications, and support their interpretations with relevant literature.
c. Figure legends should clearly explain the coding used for the actual parameters (i.e., A, B, C, D, E).
d. Given that the 2-way interactions are insignificant, the authors should explain the rationale and relevance of presenting Figure 3.
e. Line 252-253: Did the authors study the fermentation of wheat straw? Please provide references if the literature is being cited.
f. Line 256-263,340-341: One reference is cited multiple times.
g. Lines 308-310: Given that the authors observe an insignificant impact of pH on itaconic acid yield, how do they identify and define optimal pH?
h. The authors report contrasting results compared to the literature but do not adequately analyze or interpret these discrepancies in their discussion.

Additional comments

Please see the basic reporting comments. The language and presentation do not match the PeerJ standards.

---

## Round 0.2 · Major Revisions

As you can see, the reviewer still has numerous concerns and recommended rejection. I decided to give you an opportunity to revise manuscript once again. Please address all the issues pointed by the reviewer and amend manuscript accordingly.

Reviewer 2 ·

Basic reporting

In this study, the authors report biosynthesis of itaconic acid using Aspergillus niger by submerged fermentation with mango peels as the substrate. The manuscript has improved from the last time but still has prominent deficiencies in scientific content and presentation. Please see the feedback reported below.

1. Once the reviewers define an acronym, it should be consistently used in the manuscript. In line 41, IA has been defined but not utilized in lines 43 and after that.
2. Lines 54-55 and 58 convey the same meaning.
3. Lines 99-100: “Fruit peels and agricultural debris were used to make itaconic acid.” References are required to support this statement.
4. Line 129: A space between number and N is required. “0.05N H2SO4”. Line 158: A space between number and mL is required. The representation of “ml” needs to be fixed. Line 159: A space between number and g is required. Please fix this at all other places carefully in the entire manuscript.
5. Line 146: A full stop is required.
6. Line 149: “Central” should not be capitalized.
7. Line 150: “Temperature” should not be capitalized.
8. Line 175: What was the speed of stirring?
9. Line 210: RSM has already been introduced previously.
10. Line 224 has already been reported in methods.
11. Line 257: Check the spaces between words

Experimental design

1. Lines 123-126: What was the moisture content of the mango peels? What is the ideal moisture content required for this study?
2. Line 205: The authors have failed to specify whether the mango peels are on a volume, weight, or weight/volume basis even after previous comment.
3. The authors don’t rationalize their selection of parameters for CCD. Is it based on a few references?

Validity of the findings

1. Line 284-286: If the effect of temperature is insignificant, why is the itaconic acid production at lower or higher temperatures explained? Doesn’t temperature insignificance indicate that the difference observed in yields at higher and lower temperatures is “statistically insignificant”? The authors contradict their results and should instead explain why temperature is insignificant.
2. Lines 323-324: Is there an optimal pH, as the effect of pH on itaconic acid yield is insignificant? Same comment as the previous one.
3. Line 341: The authors report that agitation speed is insignificant and compare with literature but do not state why it is insignificant.

---

## Round 0.3 · accepted · Accept

In my view, all the remaining issues pointed out by the reviewer were adequately addressed and the manuscript was amended accordingly. Therefore, the revised version is acceptable now.